# Influence of Training and Single Exercise on Leptin Level and Metabolism in Obese Overweight and Normal-Weight Women of Different Age

**DOI:** 10.3390/ijerph191912168

**Published:** 2022-09-26

**Authors:** Eugenia Murawska-Ciałowicz, Agnieszka Kaczmarek, Małgorzata Kałwa, Anna Oniszczuk

**Affiliations:** 1Physiology and Biochemistry Department, Wroclaw University of Health and Sport Sciences, 51-612 Wroclaw, Poland; 2Sport Didactics Department, Wroclaw University of Health and Sport Sciences, 51-612 Wroclaw, Poland

**Keywords:** health training, leptin, women, exercise, obese, overweight, training

## Abstract

Leptin is one of the important hormones secreted by adipose tissue. It participates in the regulation of energy processes in the body through central and peripheral mechanisms. The aim of this study was to analyse the anthropological and physical performance changes during 9 month training in women of different age and body mass. The additional aim was the analysis of leptin levels in the fasting stage and after a control exercise. Obese (O), overweight (OW), and normal-weight (N) women participated in the study. Additional subgroups of premenopausal (PRE) (<50 years) and postmenopausal (POST) (50+) women were created for leptin level analysis. The main criterion of the division into subgroups was the age of menopause in the population. The control submaximal test and maximal oxygen uptake (VO_2max_) according to Astrand–Rhyming procedures was performed at baseline and after 3, 6, and 9 months. Before each control test, body weight (BM), body mass index (BMI), percentage of adipose tissue (% FAT), and mass (FAT (kg)) were measured. Moreover, before and after each test, leptin level was measured. After 9 months, there was a significant decrease in BM in the O (*p* < 0.05) and OW (*p* < 0.05) groups with no significant changes in the N group. There was a decrease in BMI in both the O (*p* < 0.05) and the OW (*p* < 0.05) groups, with no changes in the N group. The % FAT reduction was noted only in the O group (*p* < 0.05). VO_2max_ increased in each of the measured groups (*p* < 0.05). The fasting leptin level at 0, 3, 6, and 9 months were the highest in the O group. The fasting leptin level before training was highest in the O group compared to the OW group (*p* < 0.01) and the N group (*p* < 0.01). It was also higher in the OW group compared to the N group at baseline (0) (*p* < 0.01) and after 3 and 6 months (*p* < 0.01). After 9 months, the leptin concentration decreased by 20.2% in the O group, 40.7% in the OW group, and 33% in the N group. Moreover, the fasting leptin level was higher in the POST subgroup compared to the PRE group in the whole group of women (*p* < 0.05). After a single exercise, the level of leptin in the whole study group decreased (*p* < 0.05). This was clearly seen, especially in the POST group. The 9 month training had a reducing effect on the blood leptin concentration in groups O, OW, and N. This may have been a result of weight loss and the percentage of fat in the body, as well as systematically disturbed energy homeostasis.

## 1. Introduction

Research in recent years on issues related to energy metabolism has revised the general knowledge of the function played by adipose tissue, with a clear emphasis on its secretory role. Among the very many biologically active substances produced by adipocytes are compounds with different biological roles, among them hormones. One of these is leptin [1].

Leptin is referred to as the OB protein or satiety hormone. The name leptin comes from the Greek word *‘leptos’*, meaning slim. It is the first adipokine, discovered in 1994, by Friedman’s team. It is encoded by the *ob (Lep)* gene, located on chromosome 7. The molecular weight of leptin is 18 kDa, including a dissociating signal sequence. A 16 kDa mature form of the hormone is then formed, made up of a single chain of 167 amino acids [1,2].

It exerts its action through the receptors located in the tissues of various organs, indicating a very broad regulatory function. Expression of the *ob* gene is found in all adipose tissue deposits, but higher expression has been shown in subcutaneous compared to visceral tissue [2,3]. The biological activity of leptin varies and depends on the amount of the free form circulating in the blood [4]. In obese individuals, the majority of leptin circulates in free form, as a bioactive protein, and obese subjects are resistant to free leptin. In lean subjects with relatively low adipose tissue, the majority of circulating leptin is in the bound form and, thus, may not be available to brain receptors for its inhibitory effects on food intake under both normal and food deprivation states [5,6,7].

Leptin acts in the central nervous system in the hypothalamus [6,7]. Its main central function is to participate in the regulation of appetite and satiety processes—reducing cravings and inhibiting food intake. Peripherally, on the other hand, it increases lipid oxidation and energy expenditure, thereby participating in thermogenesis [8,9]. The amount of leptin produced correlates positively with adipose tissue content. Obese individuals have elevated blood leptin concentrations as a result of leptin resistance. They are also insensitive to the exogenous administration of leptin. This may be due to a variety of causes, ranging from impaired leptin transport across the blood–brain barrier to impaired receptor activation and excitatory signal propagation [10]. It regulates adipose tissue stores, controls body weight, and is involved in maintaining energy balance [10], thanks to the integration of the central nervous system with peripheral tissues involved in energy substrate utilisation processes (adipose tissue, liver, and skeletal muscle) and with the reproductive system [11,12,13]. Starvation has been found to cause a decrease in blood leptin levels and suppression of the hypothalamic–pituitary–gonadal axis [14,15].

Under conditions of energy balance, leptin is a relatively good marker of the amount of body fat. A negative energy balance leads to a decrease in leptin concentration in the blood and a positive balance leads to an increase [16,17].

As mentioned, leptin concentration is dependent on body fat [14,15]. It correlates with BMI and total body fat mass, although leptin mRNA shows greater expression in subcutaneous adipose tissue. Adipocyte size is also a determinant of leptin synthesis. Larger adipocytes of subcutaneous tissue produce more leptin compared to adipocytes of abdominal tissue [18,19].

Leptin concentration in plasma and cerebrospinal fluid is sex-dependent, being higher in women than in men in normal, overweight, and obese subjects. [7]. The differences in its blood concentration between men and women are due to body fat, BMI, and the action of sex hormones [20,21,22]. Luukkaa et al. [23] showed an inverse relationship between testosterone and leptin concentrations. Leptin concentrations may increase with age in both sexes as a result of reduced concentrations of the soluble form of the receptor. Its secretion shows a certain rhythmicity. The peak is during the night hours—around 2 am. At this time, the concentration is approximately 30–100% higher than in the morning or early afternoon [24].

In addition to a rational diet, physical exercise is the most important element of a healthy lifestyle to increase physical fitness, ensure functional independence in old age, and counteract many diseases [24]. Numerous studies confirm that a sedentary lifestyle reduces physical fitness by promoting the development of unfavourable lesions. Hypokinesia appears to be the most serious cause of disease in modern man [25].

The health benefits of systematic physical activity can be observed at various structural and functional levels of the body. From a physiological point of view, these are understood as adaptive effects. Depending on the prevention goal and the type of condition, different forms of exercise are proposed. However, the desired effects will only occur if the physical activity is framed by training and undertaken regularly, with intensity defined in terms of %HR_max_, %VO_2max_, and MET or assessed on the Borg scale, set individually, depending on the physical fitness and preferences of the persons exercising. The type of exercise and the resulting movement forms should be a consequence of the health training objective.

Although knowledge of the beneficial effects of exercise on the body at different levels of its organisation is quite extensive, little is known about the dynamics of changes in women’s blood leptin concentrations depending on body weight and age participating in the same form physical training.

Existing research findings in this area do not allow a clear answer to be formulated as to whether a single exercise session increases or decreases leptin concentrations. Opinions on the effect of physical training, including health training, on leptin concentrations are also inconclusive. Existing knowledge in this area is not systematised.

The aim of this study was to evaluate the effects of regular physical training and single exercise session on leptin levels in obese, overweight, and normal-weight women.

## 2. Material and Methods

### 2.1. Participants

In the early stage, 215 females volunteered for the study. This was far in excess of the number of women eligible for analysis. Taking into account the inclusion and exclusion criteria, 145 female volunteers qualified for the study (Figure 1). During the 9 months of training, the number of women was further reduced due to resignation from training, family reasons, too much/low load, and injuries. Women who participated in the training, but whose absenteeism during the class was greater than 10% of the class between follow-up examinations or during follow-up examinations, were also excluded from the analysis. Finally, 75 female volunteers, aged 45.55 ± 11.33 years, systematically participating in training aimed at reducing body weight and improving cardiorespiratory fitness, were selected for data analysis.

All volunteers declared good health status, which was confirmed by medical examination and written agreement of a physician. They started participation in the training to reduce their body mass and to improve their cardiorespiratory fitness. Before testing and training, all women were informed about the purpose of the tests, the procedures for performing biochemical and physical performance tests, and the possible discomfort effects after tests. A research protocol was presented to them, and safety principles were discussed. Each woman provided informed written consent to participate in the study. Moreover, they also were informed that they could resign from the study at any stage without reason given. The exclusion criteria for the study were diabetes and thyroid disease, hypertension, taking medication affecting fat metabolism, muscle and osteoarthritis pain, smoking, excessive alcohol intake, unsystematic participation in training, and follow-up examinations. All included participants declared a sedentary lifestyle. Their physical activity was less than 2 h per week. Moreover, during the training, the women did not change their current diet. However, diet was controlled in terms of quantity and quality according to the menus prepared by them. The average consumption of protein was 0.88 ± 0.18 g/kg b.w., fat 0.86 ± 0.3 g/kg b.w., carbohydrates 2.75 ± 0.64 g/kg b.w., and fibres 17.51 ± 5.38 g/day. The studies were conducted in accordance with the Declaration of Helsinki. The study was approved by the Bioethics Committee of Scientific Research at the University School of Physical Education in Wrocław, Poland (No. 14/07/2006).

Taking into account the body mass index (BMI) of the women studied, three groups were formed: obese women (O) with BMI ≥ 30.00 (age: 46.34 ± 10.59; *n* = 25); overweight women (OW) with BMI = 25.0–29.99 (age: 47.67 ± 9.86 l; *n* = 24); normal-weight women (N) with BMI < 25.00, (43.24 ± 9.56; *n* = 26). In the analysis of the results presented, results of the entire study group of women, i.e., ‘all women’ (AW), are also presented. Figure 1 shows the successive steps leading to the selection of the groups whose results were finally analysed.

Due to the large diversity in age of the women studied, two subgroups were formed within the three main groups: premenopausal (PRE) and postmenopausal (POST). The PRE groups included women aged up to 49 years, while the POST group included women aged 50+. Mean age values are shown in Table 1. The menopausal period in the Polish female population was used as the cut-off value [26]. Regardless of the age division, the premenopausal women had regular menstrual cycles and were not using hormonal contraceptives. The postmenopausal women had not experienced a menstrual cycle for at least 1 year and were not receiving hormone therapy [27]. This division was used only for the analysis of leptin concentrations. When describing leptin concentration results, groups were named using the abbreviation main group/subgroup for ease of reference. Groups were, thus, called AW/PRE, AW/POST, O/PRE, O/POST, OW/PRE, OW/POST, N/PRE, and N/POST. Major grouping was used to describe anthropometric and physical performance parameters.

### 2.2. Training Characteristics

The training lasted 9 months. The research began in October and was completed in June of the following year. The recruitment of volunteers took place according to specific criteria, including the desire to reduce body weight, age 30–65 years, and a state of health that allowed active participation in the training. Training was held twice a week. In the first training session, muscular strength, endurance, and flexibility were shaped. This was a total body condition (TBC) class. The classes comprised body weight exercises, with the introduction over time of exercises with 0.5 kg hand weights and exercises with elastic bands. Isolated exercises of individual muscle parts were performed, repeated in 1–3 series. During the second training session (Thursday), the body’s aerobic fitness was shaped. The exercises used were low-impact, cardio, fat-burning, low-intensity exercises, which are recommended by specialists precisely for obese and elderly people, as well as people with a relatively low level of physical fitness. Classes always began with a warm-up lasting 10 min, progressing to the main part, lasting approximately 45 min. The session was rounded off with calming and stretching exercises. The exercises were led by an experienced fitness instructor, specialising in health coaching classes, and accompanied by music, the tempo of which depended on the part of the workout.

Heart rate (HR) monitoring was used to assess the intensity of the training, using sportsters (Polar M400). The actual intensity of each training session was averaged across the group. The 220 − age rule was used to calculate the maximum heart rate (HR_max_) as the safety limit. Knowing HR_max_ allowed the training load to be personalised.

### 2.3. Follow-Up Examinations

Four follow-ups were performed during the 9 months of training: before training (baseline, 0 months), after 3 months, after 6 months, and after 9 months. Each follow-up examination consisted of anthropometric parameters, a fitness test, and biochemical analyses.

#### 2.3.1. Anthropometric Parameters

Each anthropometric parameter test consisted of measuring weight and height, calculating BMI, and measuring body composition. Body weight was measured with a medical scale (Radwag, Poland). Body composition was assessed using near-infrared light (NIR) spectrometry with a FUTREX-6100/XL instrument (Futrex Inc., Hagerstown, MD, USA) according to a previously described procedure [26].

#### 2.3.2. Physical Performance Submaximal Test

As the women in the study were not physically active, an exercise test, performed on a cycloergometer, was used to assess their physical performance. On the basis of the value of HR during the steady-state phase, VO_2max_ was determined using the Astrand–Rhyming submaximal test. The numerical value of this parameter illustrated the level of physical fitness of the women studied. The test consisted of a cycling on LODE’s Excalibur ergometer, with a load of 100 W. The women breathed though the face mask of a Quark b^2^ ergospirometer (Cosmed, Roma, Italy) designed for stationary testing. The load in the test was intended to be submaximal. The test was always performed when fasting, between 8:00 and 10:00 a.m.

#### 2.3.3. Biochemical Tests

Blood for biochemical determinations on each follow-up examination was collected from the antecubital vein at fasting stage, between 8:00 and 10:00 a.m. and 15 min after the exercise test.

Leptin concentration was determined by radioimmunoassay (RIA), based on the Human Leptin RIA Kit reagent kit from Linco Research LTD, USA (cat. no. HL-81K). The range of standard adopted for this method was 7.4–11.1 ng/L. Sensitivity of the method was 0.5 ng/mL. The intra-assay and inter-assay coefficients of variation were <8.3% and <6.2%, respectively.

### 2.4. Statistical Analysis

All data were statistically analysed using Statistica PL Stat Soft version 13.0 (Cracow, Poland). The data were expressed as the mean ± SEM. In all tests used, a level of *p* ≤ 0.05 was considered as statistically significant. Several tests were used to analyse the results. The normality of the distribution was tested using the Shapiro–Wilk test. If the trait was characterised by a normal distribution, one-way analysis of variance (ANOVA) was used for further calculations, preceded by Levene’s test of homogeneity of variance. If the null hypothesis of no difference was rejected (*p* = 0.05 or less), Duncan’s multiple-range test was used for further analysis. Leptin was analysed by repeated-measures two-way analysis of variance (ANOVA) followed by Bonferroni’s post hoc test for multiple comparisons. To examine the strength of the correlations between the characteristics, Spearman’s rank correlation coefficient was calculated.

## 3. Results

### 3.1. Anthropometric Parameters

All anthropometric results are presented in Table 2. Body mass in AW gradually decreased as a result of 9 months of training. Significant differences were found between the baseline value and the values obtained after 6 (*p* < 0.05) and 9 (*p* < 0.05) months of training, as well as between the examination performed after 3 months compared to the status after 9 months (*p* < 0.05). Significantly higher body mass values in the AW group were recorded in the older (postmenopausal) in comparison to younger (premenopausal) women at baseline (*p* < 0.05) and after 3 months (*p* < 0.05). In obese women, a visible effect of training on body mass reduction was found after only 3 months of training. Further significant weight loss compared to baseline was recorded after 6 (*p* < 0.05) and 9 (*p* < 0.05) months. After 9 months, weight stabilisation was observed compared to the examination after 6 months.

In the OW group, a reduction in body weight after training was also observed, although the effect relative to baseline was only found in the examination after 6 months. A further reduction was found after 9 months. In the N group of women, there was no reduction in body weight as a result of training at any of the time periods studied.

BMI values in the entire group decreased after training. Significant differences were noted between the baseline compared to the values after 9 months and between 3 and 9 months of training (*p* < 0.05). In the O group, values significantly lower compared to baseline examination were recorded after 6 and 9 months (*p* < 0.05). In the OW group, a significantly lower BMI was recorded in the examination after 6 months compared to baseline and between the baseline and after 9 months (*p* < 0.05). No statistically significant differences were found in the N group.

The fat percentage (%FAT) in the AW group did not change in any of the study periods compared to baseline. It only decreased significantly in group O after 6 months (*p* < 0.05) of training compared to baseline values. A further statistically significant reduction was observed in the examination after 9 months (*p* < 0.05). In the OW and N groups, the applied training did not result in changes in body fat percentage.

A reduction in fat mass (FAT (kg)) was recorded in the AW group after just 3 (*p* < 0.05) months of training. Significant differences from baseline were also found in the examinations at 6 and 9 months (*p* < 0.05). In group O, training resulted in a decrease in fat mass recorded in all three examinations compared to baseline. Statistically significant differences compared to the baseline examination were noted in each subsequent examination, but also between months 3 and 6 (*p* < 0.05) and between months 6 and 9 (*p* < 0.05). In the OW group, lower FAT values (kg) were observed after 6 and 9 months compared to baseline. No statistically significant differences were found in the N group.

### 3.2. Training Intensity and Physical Performance

The HR used to monitor training intensity indicated moderate loads during the warm-up; as training continued, the HR decreased to values characterising light work. Taking into account the age of the participants in the classes, it was found that the training HR was between 49% and 78% of HR_max_, indicating moderate- to heavy-intensity work.

By analysing the mean HR from the steady-state phase during the follow-up examinations in the subsequent studies, it was found across the entire group of women that the pre-training (baseline) HR was the highest with an average of 157.9 ± 13.60 bpm. In each successive examination, the HR value of the steady-state phase decreased significantly compared to the baseline but remained at all times between values reflecting heavy and very heavy work. Work intensity during testing averaged 84% HR_max_ at baseline, 81% after 3 months of training, 79% after 6 months, and 67% HR_max_ at 9 months.

As a result of the training, the physical performance of the female participants improved significantly. The lowest VO_2max_ was found at baseline. Significantly higher values for this parameter compared to baseline, and subsequent examinations were recorded after 9 months of training. Exercise intensity averaged 85%VO_2max_ in the first examination, 87% VO_2max_ in the second, 72% VO_2max_ in the third, and 63% VO_2max_ in the last. The VO_2max_ values obtained in the follow-up examinations in each of the study groups are shown in Table 2.

### 3.3. Biochemical Test

The results of fasting leptin concentrations in the subsequent examinations are shown in Figure 2. In the whole group of women, fasting leptin concentrations were reduced by training. In each subsequent follow-up examination, lower concentrations were observed compared to the earlier examination in all groups. In group O, lower leptin levels compared to baseline were found at 3, 6, and 9 months. No statistically significant differences were recorded between examination at 6 and 9 months. In the OW group, a reduction in leptin levels was only recorded in the examination after 6 and 9 months. These values were lower compared to the baseline. No differences were observed between 3 and 6 months of training.

In the N group, lower leptin concentrations compared to the OW group at baseline were observed only in the examination after 6 months of training. A further decrease in leptin levels compared to baseline and after 3 months was found in the examination at 9 months. However, it was not statistically significant compared to the examination at 6 months.

In each examination, the highest leptin values were recorded in group O. These differed significantly from leptin concentrations in group OW (*p* < 0.01) and group N (*p* < 0.01). Significantly higher values were also observed in group OW compared to group N, but only in examinations at baseline and after 3 and 6 months.

Fasting leptin levels were reduced by 20.2% in the O group of women after 9 months of training, by 40.7% in the OW group, and by 33% in the N group.

Analysing fasting leptin concentrations in the AW, divided into PRE and POST (Figure 3), it was found that older women (POST) had significantly higher leptin values in each examination (*p* < 0.01) in comparison to the PRE group. The same differences were found in the O group although statistically significant differences were recorded only in the examination after 6 (*p* < 0.05) and 9 (*p* < 0.05) months. In the group of OW, higher concentrations in POST women were recorded at 0, 3 and at 9 months, while, in the group of N women, they were recorded at 0, 3, 6 and 9 months.

Analysing leptin levels after a single exercise (Figure 4), there was a statistically significant decrease in leptin levels in the AW (*p* < 0.05), O (*p* < 0.01), OW (*p* < 0.05), and N (*p* < 0.05) groups in all examinations.

The largest decreases in concentration were seen in the obese female group: at baseline by 31.4%, after 3 months by 31.1.%, after 6 months by 39%, and after 9 months by 29.5%. For the OW group, concentration values decreased by 23% at baseline, by 15% after 3 months, by 7% after 6 months, and by 37% after 9 months. In the normal-weight group, the decrease was 13.9%, 11.5%, 18.9%, and 17%, respectively.

Considering the age distribution, it was found that leptin concentrations decreased after exercise in both PRE and POST groups (Figure 5).

In the AW group of premenopausal women (AW/PRE), lower post-exercise concentrations were observed in all examinations than in the POST group (*p* < 0.05) with the exception of the third examination.

In the O/PRE group, significantly lower post-exercise leptin concentrations persisted into the examination after 6 months. In the fourth examination, exercise no longer affected its concentration. In the O/POST group, exercise significantly reduced leptin concentrations in all tests. In the OW group, an effect of effort in both PRE and POST was observed only at baseline and after 9 months. In the N/PRE group, a significant decrease in its concentration was recorded only at baseline (*p* < 0.05) and after 6 months (*p* < 0.05), while in the N/POST women, it was recorded in the examinations after 3 (*p* < 0.05) and 9 (*p* < 0.05) months.

When testing for correlations between leptin and anthropometric and performance parameters, a correlation with age was found in the entire group of women studied, in all study periods. In the subsequent study periods, the correlation coefficient was *r* = 0.32, *r* = 0.43, *r* = 0.53, and *r* = 0.63, along with a negative correlation with VO_2max_ (*r* = −0.56; *r* = −0.48; *r* = −0.72; *r* = −0. 56), and a positive correlation with body weight (*r* = 0.58; *r* = 0.43; *r* = 0.66; *r* = 0.54), BMI (*r* = 0.67; *r* = 0.67; *r* = 0.50; *r* = 0.69), and %FAT (*r* = 0.62; *r* = 0.67; *r* = 0.50; *r* = 0.46).

## 4. Discussion

Assessing the regulatory role of exercise in relation to adipokines appears to be an important health issue in the context of metabolic disorders of the body and their prevalence in developed societies. The homeostatic role of these substances is very important. Physical exercise appears to be a valuable tool in regulating the metabolic function of adipose tissue and, therefore, the hormones it secretes [28]. However, a similar direction of response should not be sought at all costs in people with different body weights, due to their body fat content and metabolic and hormonal activity [28,29].

In the current study, the highest fasting leptin concentrations were recorded in O women, with the lowest in N women. In the present study, fasting leptin concentrations were significantly reduced in each of the study groups following 9 months of training. The first training effects in this area were recorded in O women as early as 3 months after starting the classes. In women in group N, although no change in body weight was observed, a reduction in leptin levels was noted. Physical fitness also improved. An increase in VO_2max_ of 44.84% was observed in the O group, with 16% in the OW women and 16.7% in the N group. Such a marked improvement in aerobic fitness particularly evident in O women is likely to have occurred as a result of a significant reduction in body weight and fat percentage.

Hickey et al. [28] observed a similar correlation. After 12 weeks of aerobic training of normal-weight, previously physically inactive women, they observed a reduction in leptin concentrations, despite no change in body weight.

Slightly different conclusions were drawn from the study by Kraemer et al. [30]. After 9 weeks of aerobic training in obese postmenopausal women, a 12.29% increase in VO_2max_ was found. However, there were no changes in leptin concentrations or reductions in BMI and fat content.

Desgorces et al. [31] assessed changes in leptin and insulin levels at 2 h and 24 h post exercise after 36 weeks of training rowers, during which they performed two follow-up examinations lasting 90 min at an intensity of 70–75% VO_2max_. Insulin concentrations decreased markedly 2 h after exercise compared to the pre-exercise value. Leptin concentrations also decreased, but the observed post-exercise effect was delayed, persisting even after 24 h. The authors found no decrease in the resting value of this hormone as a result of training, although it correlated with body weight and fat content.

The same team in another study found a high negative correlation between leptin and norepinephrine concentrations in the second hour of restitution. At the same time, adrenaline and dopamine concentrations were also relatively high. The authors concluded that norepinephrine may be a regulator of leptin concentrations during the restitution period, as, during exercise, the stimulation of catecholamine secretion results in changes in body fat [32]. In these studies, very high concentrations of free fatty acids (FFAs), which still persisted after 120 min, were also observed, which were also considered by some authors to be a leptin-lowering factor after exercise.

In our study, leptin levels correlated with body weight, BMI, and fat content. There was a marked improvement in cardiorespiratory fitness across the group. A negative correlation of leptin with VO_2 max_ was found in the entire study group of women, irrespective of body weight in each study period. It seems that it is not only the negative energy balance achieved during exercise or the degree of nervous system excitation and the secretion of catecholamines that are determinants of post-exercise changes in leptin concentrations and changes in resting leptin values as a result of training, but also the very broad adaptation observed in various functional areas.

The decrease in resting blood leptin levels as a result of training observed in our study was also found in studies by other authors.

Ramazan et al. [33], after 4 weeks of training in obese women with an average BMI = 40.7 kg/m^2^, obtained a significant decrease in leptin concentrations despite no reduction in body weight. The training consisted of a daily workout of 45 min, at an intensity of 60–80% HR_max_. A similar training intensity was recorded in our study.

In our study, a significant decrease in leptin levels in group O was observed after only 3 months of training, as well as after 6 months in groups OW and N, although no reduction in body weight was recorded at all in group N. These dynamics of change lead to the conclusion that the greatest training changes, and the fastest observed, occur in those with the highest body fat content. The changes in leptin concentrations in the observations presented here coincide with a significant decrease in body weight in O women after only 3 months. The number of training sessions also seems to play a role. Ozcelik et al. [34] observed a 41.41% decrease in fasting leptin after 12 weeks of endurance training.

In our study, in a group of obese women, we recorded a decrease in concentration of 14.56% after 3 months and 20.2% after training. In our study, it was only two sessions per week, while Ozelik et al. [34] used 3–4 sessions. In the study of Ramazan et al. [33] exercise sessions were held daily, indicating that training effects are dependent on the frequency of activities. With a lower frequency of classes, a longer time is needed to achieve similar results.

In a study by Charmas et al. [35] conducted in middle-aged women participating in a single aerobics class held at an intensity of 70% HR_max_ for 60 min, leptin concentrations measured after the class and at 12 h restitution showed a decreasing trend, while FFA concentrations were significantly increased and remained at this level until the following day. This prolonged use of energy from the breakdown of triglycerides may contribute to weight loss and a decrease in body fat following systematic aerobic exercise. Long-term elevated concentrations of FFAs may act in the negative feedback loop, inhibiting central signals that stimulate leptin secretion, which in turn can be observed as a reduction in leptin concentration after exercise and lower fasting leptin secretion.

Studies by other authors have found that, with a decrease in leptin, there is a marked increase in IGF-1 concentrations immediately after exercise, which decreases 60–90 min after exercise. Presumably, the decrease in leptin is due to an increase in IGF-1 [36]. IGF-1 is a GH-dependent growth factor that plays an important role in the adaptation of skeletal muscle to exercise loads. Exercise stimulates GH secretion. This is also made possible by the effect of IL-6 on the pituitary gland. In addition, GH increases lipolysis, indirectly enhancing the adipocyte response to catecholamines [37], and its elevated levels can persist for 1–2 h of restitution [38].

The regulatory effect of IL-6 on leptin secretion by adipose tissue was also argued by Trujillo et al. [39], while Nemet et al. [40] believed that this is quite a plausible mechanism, as, during exercise, muscle contractions result in increased secretion of IL-6, which is thought to integrate and regulate hormonal and metabolic functions of the body during exercise.

In in vitro studies, Ropelle et al. [41] further found that, under the influence of exercise, leptin modulates the action of the AMPK/mTOR pathway at the level of the hypothalamus, in an IL-6 dose-dependent manner. Leptin acts antagonistically to GH. Thus, the inhibition of its secretion by GH after exercise is a very plausible mechanism, since GH concentrations are increased, while leptin concentrations, as shown in our study and those of others, are decreased.

In animal studies, leptin administration resulted in inhibition of GH secretion. This effect was due to the regulatory effect of leptin on somatostatin gene expression and a direct effect on GH gene expression [42].

The mechanism of the antagonistic action of leptin and GH could explain, at least in part, the decrease in leptin concentration in response to a single exercise session, as well as the training used in our training examinations. Growth hormone is known to be a potent stimulator of lipolysis, and the neuroregulatory secretion of GH is closely dependent on adipose tissue stores. In normal-weight people, GH secretion is higher; in obese people, GH concentrations are low [43].

This is a negative feedback effect between FFAs and GH. Elevated concentrations of FFAs characteristic of obese individuals inhibit growth hormone secretion in a reciprocal manner [43].

Such correlations could also explain why the blood leptin concentration of obese people is higher compared to normal-weight people.

The interplay between GH and leptin may be enhanced by signals transmitted by ghrelin, which is a stimulator of GH secretion and whose concentration increases after exercise [44]. In a study by Foster-Schubert et al. [45], ghrelin concentrations were also increased after 12 months of aerobic training performed by obese postmenopausal women. The authors attribute this change to the body’s adaptive response to a reduction in body fat.

According to Fagionni et al. [46], leptin plays a role in the inflammatory response via stimulation of the immune system. It enhances cytokines production and phagocytosis by macrophages. The authors showed that administration of lipopolysaccharides, IL-1, or TNF-α increases levels of the mRNA for leptin in adipose tissue and circulating levels of leptin. According to an experiment, they concluded that IL-1β is essential for leptin induction. These data suggest that leptin induction during inflammation is regulated in a manner similar to the cytokine response to infection and injury. This could be one of the mechanisms underlying and explaining the inflammation process development in obese people.

According to Antonioni et al. [47], plasma IL-1β and caspase-1 levels are inversely correlated with insulin resistance. They presented in patients with morbid obesity and type 2 diabetes mellitus with normal glucose tolerance that caspase-1 levels are normalised after weight loss, whereas IL-1β is normalised only in people without diabetes mellitus. This suggests the persistence of a systemic inflammatory condition in people with this disease.

An attempt to explain the changes in leptin observed in our study after a single exercise session and after training, as a function of the existence of reciprocal correlations among leptin, GH, and ghrelin, is interesting and quite feasible, as ghrelin has an inhibitory effect on leptin secretion. It is a hormone that stimulates craving.

In the present study, leptin concentrations immediately after exercise were reduced in each of the women’s groups and in each study. The greatest reduction in leptin concentration was observed in obese women, which may indicate that leptin secretion is significantly inhibited under exercise stimuli, especially in obese individuals.

In a study by Murawska-Ciałowicz et al. [48], in a group of cyclists (men and women) subjected to a progressive test on a cycloergometer, a decrease in leptin concentration after exercise was also observed, but only in women. In contrast, throughout the group, it was found to be positively correlated with BMI and negatively correlated with the amount of energy expended during the test.

Kraemer et al. [30] in postmenopausal women using and not using hormone replacement therapy also observed a reduction in leptin levels after 30 min of exercise at 80% VO_2max_ intensity. This was accompanied by increases in cortisol and GH, which regulate leptin secretion.

Some authors believed that efforts of up to 60 min, regardless of intensity, may not modulate leptin secretion [49], which contradicts the results presented in the current paper and the work of other authors.

In the current study, the exercise lasted only 10 min, and a significant reduction in leptin concentration was found. Thus, it appears that it is not the time criterion that is the determinant of leptin changes in the blood. Perhaps it is the intensity of the provocation of anaerobic metabolism and rapid glucose consumption. It also appears to be important whether the subjects are fasting or have eaten a meal when starting the exercise.

The women participating in the study presented in this paper were fasting when taking the 10 min test, and the test was always performed in the morning between 8:00 and 10:00 a.m. Hilton and Loucks [50] believed that the decrease in leptin concentration only occurs when exercise was performed on an empty stomach, which creates conditions of greater negative energy balance, limiting leptin secretion to a greater extent.

In every group of women we studied, a significant decrease in leptin concentration was observed at 15 min of restitution after exercise with a 100 W load. The effort provoked an increase in HR to values indicative of heavy work and was performed under anaerobic conditions, which was reflected in lactic acid concentrations exceeding the threshold for anaerobic metabolism. Although the effort was hard, the energy expenditure did not exceed 100 kcal. It is also important to note that the decrease in leptin levels was much greater in the O women compared to the N group, which was probably related to higher energy expenditure.

The nature of changes in leptin concentrations depends on a number of concomitant factors (changes in cortisol, insulin, and glucose) stimulating leptin secretion and adrenaline and noradrenaline, or GH inhibiting its secretion. Post-exercise changes may also depend on diurnal rhythm, as well as changes in plasma osmolarity [51], and are not observed if energy expenditure during exercise does not exceed 800 kcal [30,49,52].

However, the conclusions of the cited authors regarding the amount of energy expenditure cannot be considered the only condition for reducing the concentration of this hormone immediately after exercise, although it is certainly important. In the current study, however, leptin concentrations were shown to decrease after exercise with a much lower energy cost than suggested by the aforementioned authors.

Mueller et al. [53] conducting studies on isolated rat adipocytes observed that leptin secretion was directly proportional to glucose consumption. When cellular glucose uptake was blocked, leptin secretion was inhibited in a dose-dependent manner.

This observation seems rather interesting and may partly explain the post-exercise reduction in blood leptin concentration, as post-exercise restitution is a state characterised by a decrease in blood glucose concentration, used during exercise as an energy source. A study by Couillard et al. [54] analysing blood leptin concentrations in men and women in the context of its association with metabolic risk factors for cardiovascular disease showed that there is a high positive correlation between fasting and post-glucose blood glucose and insulin concentrations and leptin concentrations. Similar observations were made by Boden et al. [55] in obese and normal-weight patients subjected to 52 h of fasting. In this study, leptin levels fell by 72% in the O and by 64% in the N, showing high correlations with insulin and blood glucose levels. Both substances, therefore, appear to be involved in the regulation of leptin secretion.

According to some authors, leptin concentrations do not change immediately after exercise. However, a delayed reduction in concentration is found. Torjman et al. [51] observed such changes 240 min after maximal exercise on a moving treadmill (a decrease of 7%) and after prolonged exercise at an intensity of 50% VO_2max_ (a decrease of 9%), also indicating the existence of a positive correlation between leptin and glucose concentration and body fat, as well as a negative one with resting energy expenditure.

Nindl et al. [56], in line with the study presented here, found a reduction in leptin concentrations after very heavy exercise. Some authors believe that prolonged exercise also reduces blood leptin concentrations. Such changes were observed by Vatansever-Ozen et al. [57] in male trainees after running on a moving treadmill at an intensity of 50% VO_2max_ for 60 min. The authors noted a 17% decrease in leptin concentrations and a 45% increase in acylated ghrelin concentrations, explaining these changes by the involvement of ghrelin in the regulation of appetite processes occurring as a result of a negative energy balance, resulting in a post-workout decrease in leptin concentrations. Trembely et al. [58] stated that reduced leptin concentrations persist for a long time after endurance exercises. Similar effects were observed by Zaman et al. [59].

In a study by Karamouzis et al. [60] following a 25 km swimming marathon, leptin concentrations decreased in all participants immediately after the event. There was an 81% increase in neuropeptide Y concentration, glycerol concentration, and FFAs. They observed a high negative correlation between leptin and neuropeptide Y, as well as between leptin and glycerol. The changes that occurred were explained primarily as the result of compensation for a negative energy balance during prolonged exercise.

A similar direction of change was observed by Zaccaria et al. [61] after an ultramarathon in trained runners (energy expenditure of 7000 kcal) with no change in the half marathon (expenditure of 1400 kcal), as well as by Leal-Cerro [62] in long-distance runners after a marathon (energy expenditure of approximately 2800 kcal). In Zaccari [61], leptin concentration correlated highly negatively with post-exercise FFA concentration and with body weight and resting fat content.

## 5. Conclusions

On the basis of our study, it can be concluded that the fasting level of leptin in women’s blood is dependent on body mass and body fat content, as well as on age. Women after menopause have a much higher leptin level compared to those women who have not yet reached menopausal age.

Training lasting 9 months reduces the leptin levels. This effect of training seems to be not dependent on body mass because the effect of training on leptin level was observed in obese, overweight, and normal-weight women.

Reductions in body weight and body fat percentage, as well as systematically disturbed body homeostasis, appear to be the main reasons for the decline in leptin levels in women of different ages and BMIs.

## Figures and Tables

**Figure 1 ijerph-19-12168-f001:**
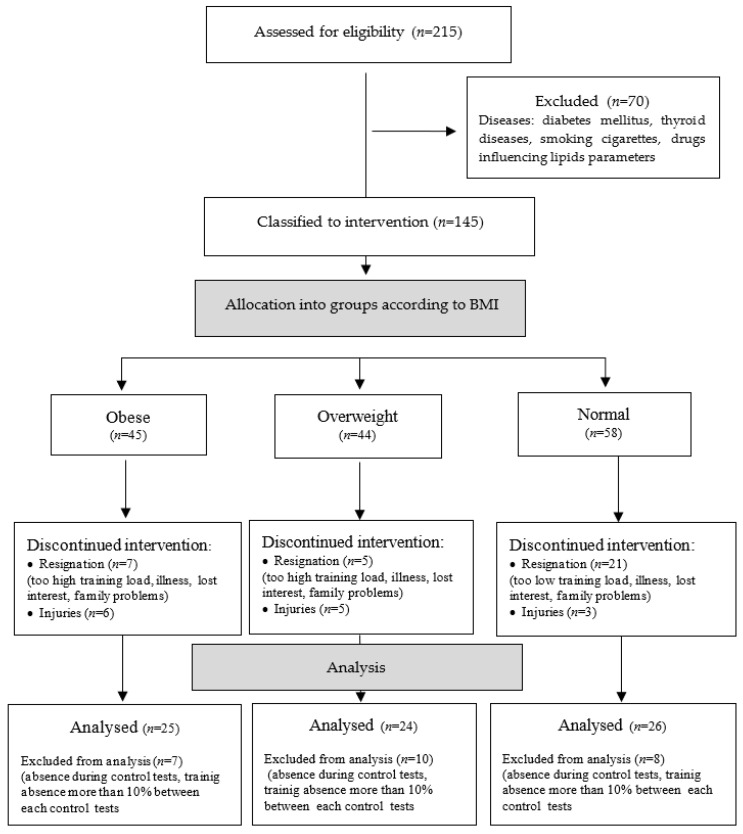
Flow diagram of participant allocation.

**Figure 2 ijerph-19-12168-f002:**
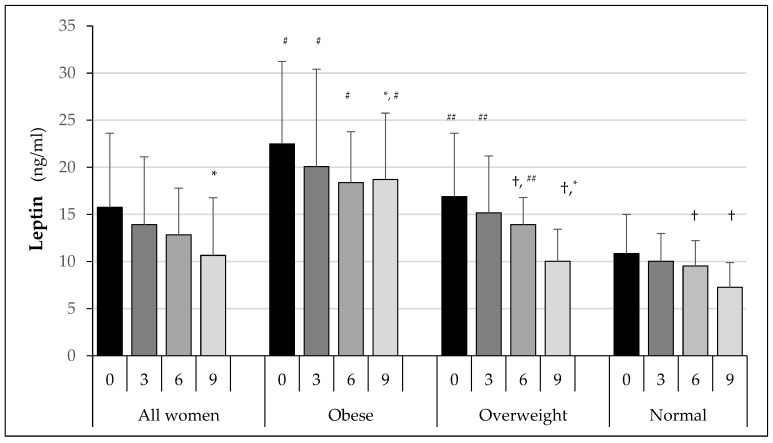
Changes in fasting leptin concentrations in the study groups over successive study periods. * *p* < 0.05 in comparison to previous months; ^#^
*p* < 0.01 in comparison to OW and N groups; ^##^
*p* < 0.01 in comparison to N group; ^†^
*p* < 0.05 in comparison to baseline.

**Figure 3 ijerph-19-12168-f003:**
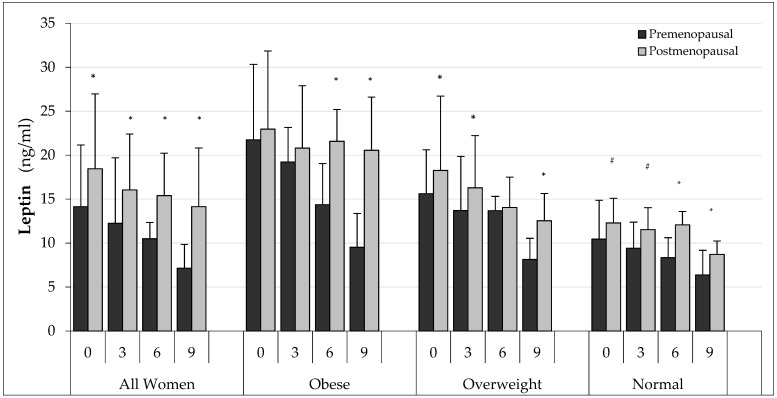
Changes in fasting leptin concentrations in the study groups by PRE and POST. * *p* < 0.01 in comparison to premenopausal women; ^#^ *p* < 0.05 in comparison to premenopausal women.

**Figure 4 ijerph-19-12168-f004:**
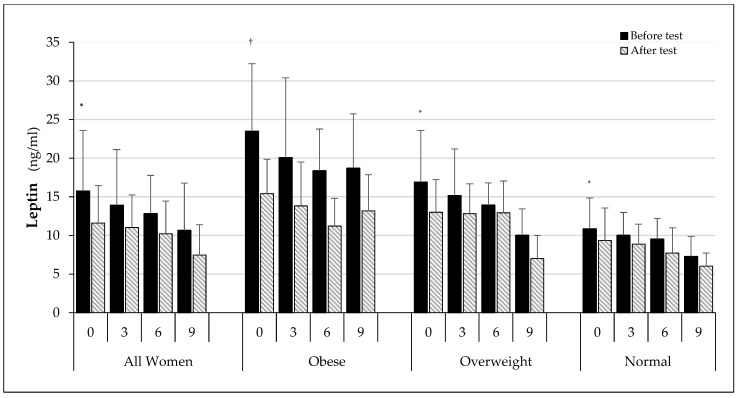
Fasting and post-exercise leptin concentrations in subsequent examinations. † *p* < 0.01 in comparison to values after test in all stages of O group; * *p* < 0.05 in comparison to values after test in AW, OW, and N groups in all stages.

**Figure 5 ijerph-19-12168-f005:**
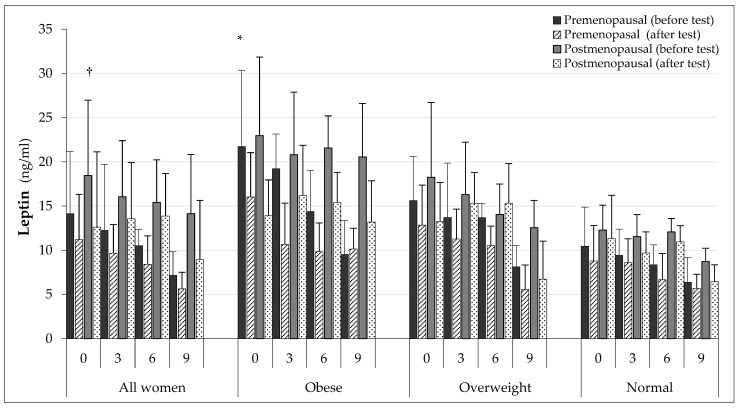
Fasting and post-exercise leptin concentrations in groups of younger and older women. * *p* < 0.05 in comparison to value after test in all PRE in all groups and months except after 9 months in O group and after 3 and 9 months in N group; † *p* < 0.05 in comparison to value after test in all POST in AW, O, OW, and N group except after 3 and 6 months in OW group, and at baseline and after 6 months in N group.

**Table 1 ijerph-19-12168-t001:** Age of women in the study groups after division into pre- and postmenopausal groups.

Group	All Women (AW)	Obese (O)	Overweight (OW)	Normal (N)
Premenopausal (PRE)	38.24 ± 8.32 (*n* = 42)	38.07 ± 7,74 (*n* = 12)	40.05 ± 8,36 (*n* = 13)	37.81 ± 8.30 (*n* = 17)
Postmenopausal (POST)	56.67 ± 5.58 (*n* = 33)	55.26 ± 4.58 (*n* = 13)	57.19 ± 5.54 (*n* = 11)	54.93 ± 4.39 (*n* = 9)

**Table 2 ijerph-19-12168-t002:** Anthropometrical and physiological parameters in all groups and measured stages.

Parameter	Stage	All Women (AW)	Obese (O)	Overweight (OW)	Normal (N)
	0	70.51 ± 12.73 ^#^	86.67 ± 12.15 ^†,††^	71.79 ± 6.12 ^##^	60.68 ± 5.36
Body mass	3	70.06 ± 10.09 ^##^	83.81 ± 10.15 ^#^	70.23 ± 5.61	62.18 ± 4.01
(kg)	6	68.34 ± 8.99	79.97 ± 7.08 ^##^	69.83 ± 5.61	61.86 ± 4.13
	9	67.33 ± 9.56	76.47 ± 6.36	68.10 ± 5.13	59.67 ± 5.67
	0	26.49 ± 4.49 ^##^	32.67 ± 2.43 ^#^	26.98 ± 1.46 ^#^	22.68 ± 1.78
BMI	3	26.43 ± 3.63 ^##^	31.59 ± 2.81 ^#^	26.97 ± 1.29	23.08 ± 1.83
(kg/m^2^)	6	25.69 ± 3.88	30.83 ± 2.44	25.75 ± 1.68	22.58 ± 1.86
	9	25.50 ± 3.66	29.54 ± 2.17	24.46 ± 2.67	22.35 ± 1.85
	0	34.63 ± 5.82	41.35 ± 3.28 ^#^	35.75 ± 2.10	28.98 ± 3.78
FAT	3	34.34 ± 5.25	40.93 ± 4.53 ^##^	35.28 ± 2.86	30.13 ± 3.82
(%)	6	33.75 ± 5.64	40.21 ± 3.81	35.36 ± 3.99	29.66 ± 3.62
	9	33.78 ± 5.46	38.58 ± 3.77	35.47 ± 3.97	29.36 ± 3.95
	0	25.13 ± 8.58 ^†^	35.82 ± 7.07 ^†^	25.59 ± 4.53 ^#^	17.84 ± 3.38
FAT	3	24.30 ± 6.66	34.30 ± 6.49 ^#^	24.79 ± 2.79	18.84 ± 3.00
(kg)	6	23.72 ± 6.72	32.60 ± 5.06	24.49 ± 3.64	18.60 ± 3.46
	9	23.15 ± 6.52 ^††^	29.53 ± 4.44	24.18 ± 5.26	17.80 ± 3.80
	0	157.90 ± 13.60 ^†^	155.30 ± 14.35 ^†^	158.20 ± 13.65 ^†^	158.90 ± 13.22 ^†^
HR	3	152.50 ± 13.91	148.10 ± 12.37	153.80 ± 13.68	152.20 ± 14.57 ^##^
(b/min)	6	148.80 ± 14.83	145.60 ± 12.25	146.50 ± 11.66	151.50 ± 17.19
	9	146.60± 14.30	141.40 ± 13.55	146.70 ± 9.55	148.10 ± 16.62
	0	27.04 ± 7.94	20.87 ± 5.79	25.01 ± 4.96	31.81 ± 7.97 *
VO_2max_	3	28.60 ± 8.20	23.63 ± 5.28	27.06 ± 6.38	33.32 ± 8.85 **
(ml/kg/min)	6	31.65 ± 9.75	26.01 ± 5.65	29.13 ± 5.86 ^###^	34.55 ± 10.82 ***
	9	33.49 ± 9.41 ^†††^	30.23 ± 5.59 ^†††^	31.04 ± 8.72 ^†††^	37.11 ± 10.46 ^†††^

^#^ *p* < 0.05 in comparison to stages 6 and 9; ^##^ *p* < 0.05 in comparison to stage 9; ^###^
*p* < 0.05 in comparison to baseline; ^†^
*p*< 0.05 in comparison to stages 3, 6, and 9; ^††^
*p* < 0.05 in comparison to stage 3; ^†††^
*p* < 0.05 in comparison to all previous stages; *, **, *** *p* < 0.05 in comparison to O and OW groups in the same control stage.

## Data Availability

The data are available on request from the corresponding author.

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
