# Peer review of "Influence of Training and Single Exercise on Leptin Level and Metabolism in Obese Overweight and Normal-Weight Women of Different Age"

_ijerph, 2022, doi:10.3390/ijerph191912168_

Round 1

Reviewer 1 Report

The study evaluates the effect of training and single exercise session on endogenous concentration of leptin in different population of obese, overweight and normal weight women.  The study is novel and interesting, well written, the results are clary presented. The discussion goes straight to the point. The reduction in leptin concentration, particularly of the Obese groups of women, thanks to the physical activity could open the way to new weight reduced strategy. In fact, as is clary state in the manuscript, Obese people are leptin resistant and this is a limit for using leptin and leptin receptor antagonist in therapy for weight loss. The possibility by engaging this type of protocol to became ’responsive’ to the leptin signal, by changing the leptin level is of note. The manuscript deserves to be published.

The eating habits and lifestyle information were register by a food diary? I suggest to add this information in the material and method section.

Author Response

Dear Reviewer,

Thank you very much for your efforts put into improving  our manuscript. We have implemented the information about the nutritional hebits and lifestyle into the manuscript. All correction are made by yellow.

Author Response

Dear Reviewer,

Thank you for your efforts put into improving  our manuscript.. We tried to addressed all your comments and suggestion.

We have made all the changes you have suggested,

In the introduction section we have refreshed the text treated about leptin bioevailability.

The references have be checked and corrected.

Typos corrected.

We have also discuss with our expert the statistics methods used and he decided that we can leave the results in such form as it was previously. 

Significance marks incorporated

In the discusion section incorporated  text based on articles suggested, trated about leptin role in immunology system and inflammation.

Thank you once more for your suggestions and comments. We strongly believe that your hard work has made our manuscript more clear.

Reviewer 3 Report

 I believe the authors tap into an important topic in physical activity and health in adult women populations.

This study aims to examine the influence of training and single exercise on leptin level and metabolism in obese overweight and normal-weight women of different ages. I think some things need clarifying for the publication that will help the overall interpretation and understanding of the results before being published within the scope of IJERPH.

Comment 1: The authors have presented a clear and well-written manuscript.

Material and Methods

Comment 2: The criteria do identify the pre and postmenopausal groups were the age? The criteria for inclusion or exclusion for the groups should be: "the premenopausal women had regular menstrual cycles and were not using hormonal contraceptives. The postmenopausal women had not experienced a menstrual cycle for at least 1 yr and were not receiving hormone therapy." https://journals.physiology.org/doi/full/10.1152/japplphysiol.00527.2018?rfr_dat=cr_pub++0pubmed&url_ver=Z39.88-2003&rfr_id=ori%3Arid%3Acrossref.org); or "(1) postmenopausal (absence of a menstrual cycle for at least one year and follicle-stimulating hormone level of >30 IU/L) and at least 40 (premenopausal) and 50 (postmenopausal) years of age on the date of the assessment". (https://www.mdpi.com/2075-4426/11/9/874/htm). 

I suggest that the authors, reclassify the groups and analyse the data based on new criteria.

Author Response

Dear Reviewer,

Thank you for your suggestions. Of course the premenopausal stage is recognised by regular menstrual cycles as well as postmenopausal stage based on the absence of menstrual cycle for at least 1 year. It is obvious for us and such cryterion has been used by us. But we wanted also underline the age of menopause in our population and overlooked to insert the main essential clinical criterion. Two goups within the three main gropus was created based on such clinical criterion. We have incorporation proper text into the body of manuscript. Thank you once more for you such essential comments .

Round 2

Reviewer 2 Report

Although not all my queries have been met, the paper has improved in light of the changes and I consider It suitable for publication.

Author Response

Dear Reviewer,

Thank you for your kindness and very merit comments.

We hav tried to make changes according to your suggestion and we think almas all of them have been addressed.

We are stongly convinced that your suggestions and comments our manuscript is now much more transparent.

Thank you,

Authors  

Reviewer 3 Report

I do not have further questions.

Author Response

Dear Reviewer,

Thank you for your kindness and very merit comments.

We have tried to make changes according to your suggestion and we think  all of them have been addressed.

We are stongly convinced that your suggestions and comments improved our manuscript which is now much more transparent.

Thank you once more,

Authors